# COPO: CONSISTENCY-AWARE POLICY OPTIMIZATION

## ABSTRACT

Reinforcement learning has significantly enhanced the reasoning capabilities of Large Language Models (LLMs) in complex problem-solving tasks. Recently, the introduction of DeepSeek R1 has inspired a surge of interest in leveraging rule-based rewards as a low-cost alternative for computing advantage functions and guiding policy optimization. However, a common challenge observed across many replication and extension efforts is that when multiple sampled responses under a single prompt converge to identical outcomes, whether correct or incorrect, the group-based advantage degenerates to zero. This leads to vanishing gradients and renders the corresponding samples ineffective for learning, ultimately limiting training efficiency and downstream performance. To address this issue, we propose a consistency-aware policy optimization framework that introduces a structured global reward based on outcome consistency, the global loss based on it ensures that, even when model outputs show high intra-group consistency, the training process still receives meaningful learning signals, which encourages the generation of correct and self-consistent reasoning paths from a global perspective. Furthermore, we incorporate an entropy-based soft-blending mechanism that adaptively balances local advantage estimation with global optimization, enabling dynamic transitions between exploration and convergence throughout training. Our method introduces several key innovations in both reward design and optimization strategy. We validate its effectiveness through substantial performance gains on multiple mathematical reasoning benchmarks, highlighting the proposed framework's robustness and general applicability. The code for this work has been open-sourced.

## 1 INTRODUCTION

Deepseek R1 Guo et al. (2025) has demonstrated remarkable potential of Reinforcement Learning (RL) in enhancing the reasoning capabilities of Large Language Models (LLMs) Radford et al. (2018); Achiam et al. (2023); Bai et al. (2023); Touvron et al. (2023); Liu et al. (2024) when tackling complex tasks such as mathematical problem solving and code generation. Previous RL applications Song et al. (2024); Ji et al. (2023a;b) based on methods such as Proximal Policy Optimization (PPO) Schulman et al. (2017), Direct Policy Optimization (DPO) Rafailov et al. (2023), and Reinforcement Learning Hunman Feedback (RLHF) Christiano et al. (2017), which primarily focus on aligning model's responses with human preferences. To better support LLMs in the exploration and prioritization of optimal reasoning paths (Chain-of-Thought, CoT Wei et al. (2022)) during training, recent works such as Qwen2.5 Yang et al. (2024) and DeepSeek R1 have shifted their attention toward outcome-based reward mechanisms and have emphasized the potential of leveraging group-relative advantage (GRA) Shao et al. (2024a) strategies for effective policy optimization.

However, despite the remarkable practical effectiveness demonstrated by these works, a growing body of studies Yu et al. (2025); Liu et al. (2025) has revealed inherent flaws in Group-relative Policy Optimization (GRPO)-based methods. Specifically, when an objective is either too trivial or too challenging for the current policy model, the reward distribution over the model's responses tends to converge, causing most relative advantages to collapse towards zero. This leads to gradient collapse and sample wastage, hindering effective optimization of the challenging objective.

DAPO Yu et al. (2025) attempts to mitigate this problem by employing dynamic batch-size sampling to improve training efficiency and stability. Nevertheless, it fails to fundamentally address the underlying sample wastage problem.

To tackle the above challenges, we propose a novel consistency-entropy-based policy optimization framework, **COPO**, that theoretically addresses the sample wastage and gradient vanishing problem under extreme samples observed in GRPO methods. Specifically, we introduce a structured global reward based on outcome consistency and a global optimization mechanism, and we incorporate an entropy-based soft-blending mechanism that adaptively balances local advantage estimation with global optimization. We not only demonstrate the performance improvement of COPO over GRPO methods in mathematical reasoning tasks, but also conduct extensive ablation studies on various existing improvements to GRPO training schemes, aiming to provide deeper insight into GRPO–based post-training methods for this domain. Our main contributions are summarized as follows:

- We analyze the problem of advantage vanishing in GRPO and propose a global advantage formulation to extract batch-level advantage signals, thereby enabling effective utilization of data samples that would otherwise be discarded due to vanishing advantages.
- We propose a novel consistency-entropy-based policy optimization method named COPO, introducing the concept of joint optimization across both intra-group and inter-group samples to fully leverage available training data.
- We develop an entropy-aware soft-blending mechanism that adaptively balances global optimization and local optimization objectives throughout training.

## 2 PRELIMINARY

### 2.1 GROUP-RELATIVE POLICY OPTIMIZATION, GRPO

GRPO, as a policy optimization algorithm, adopts a more streamlined approach by leveraging reward-based advantage estimation. The core idea of GRPO is to eliminate the need for an additional value network by computing advantages through intra-group reward comparisons under the same input. Specifically, given an input prompt $q$, the old policy $\pi_{\theta_{\text{old}}}$ generates a set of $G$ candidate output sequences: $\mathcal{O}_q = \{o_1, o_2, \ldots, o_G\}$. These sequences are then evaluated by a task-specific reward function $r_\phi$, designed according to the optimization objective, yielding a corresponding reward set: $\{r_1, r_2, \ldots, r_G\}$. The direction of policy update is determined by the relative ranking of rewards within the group: samples receiving higher rewards than the group average are encouraged by increasing their likelihood under the policy, while those with below-average rewards are suppressed by reducing their associated policy probabilities.

From this, the advantage of GRPO is calculated as:

$$\hat{A}_i = \frac{r_i - \mu_r}{\sigma_r},\tag{1}$$

where $\mu_r = \text{mean}(\{r_i\}_{i=1}^G)$, $\sigma_r = \mu_r = \text{std}(\{r_i\}_{i=1}^G)$. Substituting the new advantage $\hat{A}_i$, group $B$, and the responses $\{o_{i=1}^G\}$ sampled by the policy model into the objective function of the PPO, we can obtain the objective function of the GRPO:

$$J_{\text{GRPO}}(\theta) = \mathbb{E}_{q, \{o_i\} \sim \pi_{\theta_{\text{old}}}} \left[ \frac{1}{G} \sum_{i=1}^G \frac{1}{|o_i|} \sum_{t=1}^{|o_i|} \min\left( \frac{\pi_\theta(o_{i,t} \mid q, o_{i,<t})}{\pi_{\theta_{\text{old}}}(o_{i,t} \mid q, o_{i,<t})} \hat{A}_i, \text{clip}(\cdot)\hat{A}_i \right) - \beta\, \mathbb{D}_{\text{KL}}[\pi_\theta \parallel \pi_{\text{ref}}] \right].\tag{2}$$

### 2.2 ADVANTAGE DEGENERATION AND GRADIENT VANISHING OF GRPO

The internal mechanism of GRPO, which relies on reward mean and variance to estimate the advantage function, exhibits an inherent fragility during training. By computing advantages based on the mean and standard deviation of rewards, GRPO encourages the model to shift its output distribution toward those that match the expectation. This training strategy inevitably leads to a gradual collapse of reward variance when a given prompt $q$ becomes either too easy or too difficult relative to the current policy $\pi_\theta$. Formally, for any group $\mathcal{O}_q$, by definition Equation 1, as $Var(r) \to 0$, we have

$std(r) \to 0$, and all $r_i \approx \bar{r}$, thus $A_i \approx 0$. As a direct consequence, the gradient of the GRPO objective vanishes: $\nabla_\theta L_{GRPO} \to 0$. The degeneration of advantages and subsequent gradient vanishing substantially reduces the contribution of affected samples to policy updates, leading to a notable decline in training efficiency. As training progresses, this phenomenon tends to intensify.

During GRPO training, we observe that challenging training examples frequently lead to all G rollout trajectories producing incorrect answers, which results in the vanishing of the advantage signal. Figure 1 presents the distribution of reward lists per prompt when training the 3B model with the GRPO method. Notably, prompts for which all sampled answers are incorrect constitute the largest proportion, accounting for 56% of the training data. When including the all-correct cases, 59.9% of the training samples exhibit zero inter-group reward variance, implying that only 40.1% of the data contribute effective advantage signals during GRPO training.

To address sparse advantage signals, DAPO uses dynamic sampling to exclude all-1 or all-0 reward samples. However, this approach leads to a significant waste of training samples, especially in the above case of small-scale LLMs, where samples with all-0 accuracy make up the majority. Given the same amount of inference data, small LLMs under the DAPO training framework discard a large portion of samples, thereby slowing down the model's performance improvement. We believe that samples with zero in-group advantage still hold value, as they can provide global perspectives on optimization directions that support the overall training of the model. Therefore, extracting the effective advantage signals from the data where advantages vanish in GRPO is crucial for further improving model performance.

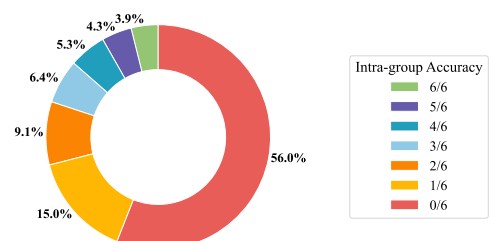

Figure 1: The distribution of intra-group accuracy for the Qwen2.5-3B-instruct model after 60 steps of GRPO training with rollout G=6. Over half of the problems yield all-zero outputs during inference.

# 3 COPO

In this paper, we proposed Consistency-Aware Policy Optimization(COPO), an RL framework that addresses the limitations of GRPO-like methods. Figure 2 shows the demonstration of COPO. To enable the effective use of samples with high consistency that would otherwise yield vanishing gradients under group-relative training, the COPO framework calculates global rewards at the batch level and yields inter-group loss. Moreover, COPO introduces a consistency-entropy-based hybrid mechanism to effectively integrate intra-group local optimization with inter-group global optimization to guide model updates.

Specifically, given a batch of prompts $Q = \{q_1, q_2, ..., q_B\}$, the training objective of COPO is defined as:

$$J_{\text{COPO}}(\theta) = \mathbb{E}_{q \sim \mathcal{D}} \Big[ w(H_q) \cdot \mathcal{L}_{\text{local}}(q) + (1 - w(H_q)) \cdot \mathcal{L}_{\text{global}}(q) \Big], \quad (3)$$

where $\mathcal{L}_{\text{local}}$ denotes the local policy loss, $\mathcal{L}_{\text{global}}$ denotes the global policy loss and $w \in (0, 1)$ is an entropy-based blending weight that adjusts the relative importance of two optimization. In the following subsections, we will describe each component of COPO in detail.

## 3.1 Intra-group local Optimization

As shown in the upper part of Figure 2, the intra-group local optimization approach follows the principles of GRPO, where rewards and advantages are computed based on responses to one prompt. For each generated response, the local reward is calculated by the rule-based reward function $R(\cdot)$ mentioned in Equation 13. The local optimization objective is expressed as:

$$J_{\text{local}}(\theta) = \mathbb{E}_{q, \{o_i\} \sim \pi_{\theta_{\text{old}}}} \left[ \frac{1}{\sum_{i=1}^{G} |o_i|} \sum_{i=1}^{G} \sum_{t=1}^{|o_i|} \min \left( \frac{\pi_\theta(o_{i,t} \mid q, o_{i,<t})}{\pi_{\theta_{\text{old}}}(o_{i,t} \mid q, o_{i,<t})} \hat{A}_{o_i}^{\text{local}}, \text{clip}(\cdot) \hat{A}_{o_i}^{\text{local}} \right) \right], \quad (4)$$

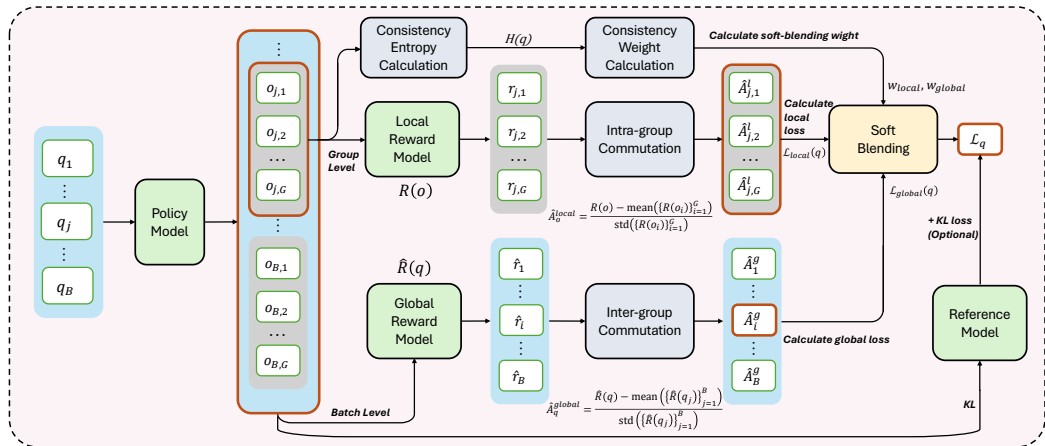

Figure 2: Demonstration of our COPO methods. COPO incorporates global optimization into the GRPO foundation to mitigate gradient vanishing caused by intra-group consistency.

where

$$\hat{A}_o^{\text{local}} = \frac{R(o) - \text{mean}\big(\{\,R(o_i)\,\}_{i=1}^G\big)}{\text{std}\big(\{\,\hat{R}(o_i)\,\}_{i=1}^G\big)} \tag{5}$$

## 3.2 INTER-GROUP GLOBAL OPTIMIZATION

When reasoning outcomes are consistently all correct or all incorrect, the group-relative local objective collapses, causing vanishing advantages and ineffective gradients. To address this, we introduce inter-prompt global optimization, using cross-prompt reward variability to drive updates even when local signals vanish.

Given a prompt $q$, we sample $G$ responses $o_{1:G} \sim \pi_\theta(\cdot \mid q)$, and define a prompt-level reward function $\hat{R}(q)$. Our goal is to optimize the policy such that it increases the likelihood of all sampled tokens in proportion to the prompt-level reward.

Under the framework of Proximal Policy Optimization(PPO), our objective remains to maximize the expected return of all sampled tokens, which is the same as intra-group local optimization. PPO calculates advantages based on Generalized Advantage Estimation (GAE), while advantage functions in traditional RL are typically computed as: $A(s_t, a_t) = G_t - V(s_t)$, where $G_t$ denotes the cumulative return from timestep $t$, and $V(s_t)$ is the estimated value function. Because training an additional value head is computationally expensive, we drop it and approximate $\hat{A}_i = \hat{R}(q) - b$, where we treat $\hat{R}(q) = \frac{1}{G}\sum_{i=1}^G r_i$ as the return $G_t$ to quantify the model's performance on prompt q, and use a baseline $b$ as a surrogate for the value function. A fixed constant baseline cannot track the reward shift that occurs during training. Instead, we use the mean reward of the current mini-batch as $b$: $b \approx \mathbb{E}_{q \sim \mathcal{B}}[\hat{R}(q)]$.

In order to keep the local and global gradient magnitudes close to each other and avoid oscillations or mode collapse, we apply standardization so that the way to calculate global advantage is the same as Equation 1. The global advantage is ultimately computed as:

$$\hat{A}_q^{\text{global}} = \frac{\hat{R}(q_j) - \text{mean}\big(\{\,\hat{R}(q_j)\,\}_{j=1}^B\big)}{\text{std}\big(\{\,\hat{R}(q_j)\,\}_{j=1}^B\big)}, for\ \forall\, o_i \in \mathcal{O}_q, \tag{6}$$

where $\text{mean}\big(\{\,\hat{R}(q_j)\,\}_{j=1}^B\big)$ and $\text{std}\big(\{\,\hat{R}(q_j)\,\}_{j=1}^B\big)$ are the mean and standard deviation of prompt-level rewards within the current mini-batch. The global optimization objective is expressed as:

$$J_{\text{global}}(\theta) = \mathbb{E}_{q,\{o_i\}\sim\pi_{\theta_{\text{old}}}}\left[\frac{1}{\sum_{i=1}^G |o_i|}\sum_{i=1}^G\sum_{t=1}^{|o_i|}\min\left(\frac{\pi_\theta(o_{i,t}\mid q, o_{i,<t})}{\pi_{\theta_{\text{old}}}(o_{i,t}\mid q, o_{i,<t})}\hat{A}_q^{\text{global}},\ \text{clip}(\cdot)\hat{A}_q^{\text{global}}\right)\right]. \tag{7}$$

While this formulation bears superficial resemblance to local advantage computation, the semantics are fundamentally different. Here, both $\text{mean}\big(\{\,\hat{R}(q_j)\,\}_{j=1}^{B}\big)$ and $\text{std}\big(\{\,\hat{R}(q_j)\,\}_{j=1}^{B}\big)$ are calculated from different actions and states; they can be viewed as trajectory-independent constants when the gradient is taken, which do not introduce bias in the policy gradient.

In the local case, samples $o_{1:G}$ within the same prompt $q$ share the same state, and the difference $R(o) - \text{mean}\big(\{\,R(o_i)\,\}_{i=1}^{G}\big)$ reflects a relative ranking among actions in that specific state. As a result, the gradient explicitly pushes the model to shift probability mass from less preferred incorrect responses toward higher-rewarding responses. In contrast, global optimization operates across different prompts $q_1, q_2, \ldots$, each representing a distinct state. The rewards $\hat{R}(q_j)$ are therefore not semantically comparable. The mean reward $\text{mean}\big(\{\,\hat{R}(q_j)\,\}_{j=1}^{B}\big)$ functions purely as a baseline to normalize the learning signal across diverse environments. Importantly, this does not cause the model to shift probability from actions in complex prompts toward those in simpler prompts. This is because the gradient in policy optimization still applies locally at each state-action pair $(s, a)$, and a constant baseline across prompts is treated as a variance-reducing term in the policy gradient, without altering the expected optimization direction.

Intuitively, the global advantage function evaluates the model's performance across different prompts within the same batch by assigning rewards or penalties accordingly. Although the specific prompts vary from batch to batch, the global advantage consistently provides positive reinforcement to trajectories that are more likely to yield correct answers. In cases where all rollout trajectories associated with a prompt are correct, the global advantage assigns the highest level of positive reinforcement to strengthen such paths. In contrast, it applies negative reinforcement to prompts for which all trajectories are incorrect. By supplying accuracy-based reward signals to data instances where the local advantage is zero, the global advantage alleviates the issue of sample inefficiency that arises when relying solely on local advantage.

### 3.3 Entropy-based Soft Blending

While the global optimization strategy effectively mitigates the gradient vanishing problem inherent to local group-relative methods, it could introduce a new challenge: the global optimization assigns the same advantage value, derived from the prompt-level reward, to all sampled responses $o_i \in \mathcal{O}_q$. Consequently, lower-quality responses may undesirably receive higher advantages than they inherently merit, thereby weakening the precision of credit assignment and diluting learning signals from truly optimal responses. Therefore, the global optimization is more suitable for prompts with high response consistency.

To address this trade-off, we propose adaptively selecting between local and global optimization strategies based on the consistency entropy of the current policy's responses. Formally, given the set generated responses $\mathcal{O}_q$, the set of outcomes extracted from $\mathcal{O}_q$ are defined as $q$: $T_q = \{\tau_1, \tau_2, \ldots, \tau_k\}$, where $k$ denotes the number of unique outcomes from $\mathcal{O}_q$.

we define the consistency entropy as:

$$H(q) = -\sum_{\tau \in T_q} p(\tau) \cdot \log p(\tau), \, p(\tau) = \frac{\text{count}(\tau)}{G}, \tag{8}$$

where $count(\tau)$ denotes the number of occurrences of $\tau$. The consistency entropy evaluates the consistency of the model's responses to a given prompt, serving as an indicator of the determinism in its output behavior.

To ensure all samples participate in both global and local optimization paths without discarding any sample entirely, we propose a soft-blending mechanism that smoothly interpolates between the two objectives:

$$\mathcal{L}_q = w_{\text{local}}(H(q)) \cdot \mathcal{L}_{\text{local}}(q) \, + \, w_{\text{global}}(H(q)) \cdot \mathcal{L}_{\text{global}}(q), \tag{9}$$

where the weighting functions are defined as:

$$w_{\text{local}}(H) = \sigma(\gamma(H - \rho)), \, w_{\text{global}}(H) = 1 - w_{\text{local}}(H), \tag{10}$$

with $\sigma(\cdot)$ denoting the sigmoid function for smooth interpolation, $\gamma$ as a temperature hyperparameter controlling the sharpness of transition, and $\rho$ the central entropy threshold around which the optimization focus transitions.

---

**Algorithm 1** COPO Training

---

**Require:** Policy model $\pi_\theta$, old policy $\pi_{\theta_{\text{old}}}$, local reward function $R(\cdot)$, global reward function $\hat{R}(\cdot)$, blending parameters $(\gamma, \rho)$, clip parameter $\epsilon$, batch size $B$, samples per prompt $G$
1: Initialize $\pi_\theta$ from pre-trained LM; copy $\pi_{\theta_{\text{old}}} \leftarrow \pi_\theta$
2: **while** not converged **do**
3:     Sample a batch of prompts $\{q_1, \ldots, q_B\} \sim \mathcal{D}$
4:     **for** each prompt $q$ in batch **do**
5:         Sample $G$ responses $\mathcal{O}_q = \{o_1, \ldots, o_G\} \sim \pi_{\theta_{\text{old}}}(\cdot \mid q)$
6:         Compute final answers $T_q = \{\tau_1, \tau_2, \ldots, \tau_k\}$ and entropy: (Equation 8)
7:         Compute blending weights: (Equation 10)
8:         Compute individual rewards $\{r_i\}_{i=1}^G$
9:         Compute group-level local advantage: (Equation 1)
10:        Compute batch-level global reward $\{\hat{r}_i\}_{i=1}^B$ for each prompt $q$
11:        Compute global advantage: (Equation 6)
12:        **for** each $o_i \in \mathcal{O}_q$, and token $t$ **do**
13:           Update the policy model $\pi_\theta$ by maximizing the COPO objective: (Equation 11)
14:        **end for**
15:     **end for**
16:     Aggregate losses over all tokens in the batch and update $\pi_\theta$ using gradient descent
17:     Periodically update $\pi_{\theta_{\text{old}}} \leftarrow \pi_\theta$
18: **end while**

---

Thus, when consistency entropy $H(q)$ is high, indicating high diversity in responses, the local optimization dominates, encouraging the model to differentiate and reinforce higher-quality responses within the group. Conversely, when $H(q)$ is low, indicating high response uniformity, global optimization dominates, pushing the model toward maintaining correctness and consistency across prompts. This mechanism enables each sample to adaptively determine its contribution intensity to both optimization pathways, mitigating potential pitfalls such as optimization precision loss resulting from relying solely on global optimization, and diminishing advantage and vanishing gradients caused by exclusively employing local optimization. Accordingly, the COPO training procedure follows Algorithm 1, with the overall optimization objective formulated as:

$$
\begin{aligned}
J_{\text{COPO}}(\theta) = \mathbb{E}_{q \sim \mathcal{D}} \Bigg[ & \frac{1}{\sum_{i=1}^G |o_i|} \sum_{i=1}^G \sum_{t=1}^{|o_i|} \cdot \Big( w(H_q) \cdot \min\Big( r_{i,t}^{(q)}(\theta) A_{o_i}^{\text{Local}}, \text{clip}(\cdot) A_{o_i}^{\text{Local}} \Big) \\
& + (1 - w(H_q)) \cdot \min\Big( r_{i,t}^{(q)}(\theta) \hat{A}_q^{\text{Global}}, \text{clip}(\cdot) \hat{A}_q^{\text{Global}} \Big) \Big) - \beta \, \mathbb{D}\text{KL}\left[ \pi_\theta \,\|\, \pi_{\text{ref}} \right] \Bigg],
\end{aligned}
\tag{11}
$$

where $r_{i,t}^{(q)}(\cdot) = \frac{\pi_\theta(o_{i,t}|q, o_{i,<t})}{\pi_{\theta_{\text{old}}}(o_{i,t}|q, o_{i,<t})}$. This normalized advantage is then uniformly applied to all log-probabilities associated with prompt $q$.

## 4   Training

To ensure fair comparisons, all experiments are conducted using the DAPO-MATH-17k Yu et al. (2025) dataset as the training set. Evaluation is performed on a suite of benchmarks, including MATH-500 Lightman et al. (2023), AIME 2024 Jia (2024), GSM8k Cobbe et al. (2021), and AIME 2025 Lin (2025), which together span a broad range of mathematical reasoning difficulties. A rule-based reward incorporating solely correctness-based signals is employed as the reward model. All training and testing experiments are conducted through the VERL framework Sheng et al. (2024).

We sample 512 prompts per batch, generating 6 responses each. The data are split into 32 mini-batches for gradient updates. Both Qwen2.5-Instruct-3B and 7B are trained for 60 optimization steps. We adopt the AdamW optimizer with no weight decay and a constant learning rate of $1 \times 10^{-6}$. For the PPO clipping objective, we apply an asymmetric clipping strategy, setting $\epsilon = 0.2$. The maximum length for both prompt and generated response is set to 2048 tokens. During inference, we use nucleus sampling with temperature 1.0 and top-p 1.0.

Table 1: Comparison of GRPO, DAPO and our method across MATH-500 and AIME24 datasets. We use mean@8 and maj@8 as metrics for MATH-500, and mean@64 and maj@64 for AIME24. The COPO results report the best performance.* denotes the results are reproduced by ourselves.

| Method | MATH 500 | | AIME 24 | | Mean Avg | Maj Avg |
|---|---|---|---|---|---|---|
| | mean@8 | maj@8 | mean@64 | maj@64 | | |
| Qwen2.5-Instruct 3B* | 48.35 | 56.11 | 2.45 | 8.36 | 45.75 | 53.41 |
| GRPO | 55.83 | 62.43 | **7.08** | **15.59** | 53.07 | 59.78 |
| DAPO | 55.93 | 61.81 | 5.47 | 13.74 | 53.07 | 59.09 |
| COPO (ours) | **60.38** | **65.06** | 6.67 | 14.48 | **57.34** | **62.2** |
| $\Delta$ (vs best) | +4.55 | +2.63 | -0.41 | -1.11 | +4.27 | +2.42 |
| Qwen2.5-Instruct 7B* | 58 | 61.73 | 9.38 | 14.7 | 55.25 | 59.07 |
| GRPO | 63.58 | 66.65 | 12.86 | 20.35 | 60.71 | 64.03 |
| DAPO | 62.15 | 65.76 | 11.77 | 17.94 | 59.3 | 63.05 |
| COPO (ours) | **65.8** | **69.27** | **13.85** | **21.07** | **62.86** | **66.54** |
| $\Delta$ (vs best) | +2.22 | +2.62 | +0.99 | +0.72 | +2.15 | +2.51 |

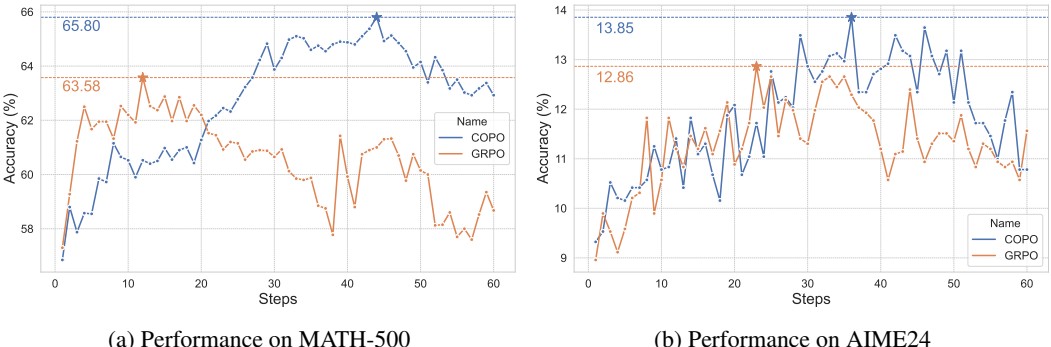

(a) Performance on MATH-500          (b) Performance on AIME24

Figure 3: Performance of GRPO and COPO on MATH-500 (mean@8) and AIME24 (mean@64) using Qwen2.5 7B Instruct during training.

For the baseline experiments, we adopt the original GRPO method without any of the enhancements introduced in DAPO, based on the experimental results presented in subsection 5.2. When applying the COPO method, we set the value of $w_{\text{local}}$ to zero for fully incorrect data, in order to prevent $w_{\text{global}} < 1$ from reducing the overall loss. More details have been depicted in the appendix.

# 5 EXPERIMENT RESULTS AND DISCUSSION

## 5.1 MAIN RESULTS

Table 1 presents the performance comparison of our proposed COPO method against GRPO and DAPO. Our method achieves superior inference accuracy over the GRPO approach with only a limited number of training steps. For Qwen2.5-Instruct 7B, COPO achieves a maximum mean@8 score of 65.8% on the MATH-500 dataset, representing a 2.22% improvement over GRPO. Moreover, COPO attains a mean@64 score of 13.85% on the AIME24 dataset, surpassing GRPO by 0.99%. In terms of the majority voting (maj) metric, COPO also demonstrates consistent improvements, achieving 69.27% (maj@8) on MATH-500 and 21.07% (maj@64) on AIME24, both outperforming the results of GRPO and DAPO.

For Qwen2.5-Instruct 3B, COPO also demonstrates impressive performance. On the MATH-500 dataset, COPO achieves a peak mean@8 accuracy of 60.38%, marking a 4.55% improvement over GRPO. When evaluated using the majority voting metric, COPO continues to show consistent gains, achieving 2.63% (maj@8) improvement over GRPO on MATH-500. However, COPO underperforms the baseline on AIME24 by 0.41%, indicating that our method cannot achieve its full potential when there is a large mismatch between model capacity and task difficulty. Additional experiments and analyses are provided in the appendix.

Figure 3 presents a comparison of the test performance of the Qwen2.5-Instruct 7B under the GRPO and COPO algorithms. Subfigures (a) and (b) show the evolution of mean@8 performance on

Table 2: Performance of different loss aggregation modes and KL divergence values on GSM8K ([†]mean@8, [‡]maj@8) and AIME25 ([†]mean@64, [‡]maj@64).

| Method | token-level loss | KL | GSM8K[†] | GSM8K[‡] | AIME25[†] | AIME25[‡] |
|---|---|---|---|---|---|---|
| | ✗ | ✓ | **86.10** | **89.56** | **3.82** | **10.00** |
| COPO* | ✓ | ✓ | 85.67 | 89.06 | 2.40 | 5.08 |
| | ✗ | ✗ | 85.62 | 88.83 | 3.02 | 8.29 |
| | ✓ | ✗ | 85.63 | 89.00 | 2.60 | 7.21 |

Table 3: Ablation study of COPO on Qwen2.5-Instruct 3B (MATH-500, [†]mean@8, [‡]maj@8). "Loss type" specifies the components of the optimization objective. "Hybrid strategy" denotes the method used to combine the local and global loss terms. "Zero control" indicates whether the local loss weight $w_{local}$ is set to 0 for samples with completely incorrect outputs.

| Method | Loss Type | Hybrid Strategy | Zero Control | MATH-500[†] | MATH-500[‡] |
|---|---|---|---|---|---|
| baseline | local | - | ✗ | 55.83 | 62.43 |
| +GO-Selective | local & global | binary | ✓ | 58.88 | 64.51 |
| +GO-Blended | local & global | soft blending | ✗ | 59.80 | 64.32 |
| +GO-Only | global | - | ✗ | 60.35 | 64.60 |

MATH-500 and AIME24 during training. As shown, GRPO achieves a rapid accuracy increase in the early stages but suffers from a performance drop in later steps. In contrast, COPO maintains relatively stable performance and achieves the best results in later training stages. This suggests that COPO, by introducing inter-group rewards and a dynamic weighting strategy, is able to extract meaningful learning signals from data with high intra-group consistency, thereby mitigating the impact of vanishing gradients caused by the zero advantage of some groups.

Notably, DAPO performs poorly compared to GRPO when trained with the same amount of data, achieving a maximum accuracy of only 5.47% on the AIME24 data set. On the 7B model, DAPO performs even worse, with its weighted mean@8 score decreasing by 1.39% relative to GRPO. These results suggest that DAPO's advantages may not be effectively demonstrated on smaller models when reasoning and training are conducted with equivalent data volumes.

## 5.2 ANALYSIS OF COPO

**Ablation Study on Implementation Modifications** We initially investigate two common modifications to the GRPO framework: token-level loss and the KL term, aiming to establish a stronger experimental baseline. We extend GRPO with global optimization by adding a global loss signal for groups with zero advantage, and then evaluate different combinations of token-level loss and KL regularization, similar to DAPO. Table 2 reports results for Qwen2.5-Instruct 3B on GSM8K and AIME25. The model performs best without token-level loss but with KL regularization, improving mean scores by 0.47% on GSM8K and 1.42% on AIME25 compared to the opposite setting. Based on these findings, we retain the original GRPO configuration for subsequent COPO optimization without additional modifications.

To demonstrate the effectiveness of different modules of COPO, we investigate three key questions. **First**, we examine whether data with zero in-group advantage truly lacks learning value. **Second**, we explore whether utilizing the global optimization can improve performance. **Third**, we aim to determine how to balance global and local rewards to maximize the model's capacity.

For the first question, we introduce the variant of GO-Selective (Global Optimization Selective), where the global optimization is applied to a prompt only when all of the extracted answers of this prompt are incorrect, and in all other cases, the local reward from GRPO is used without modification. For the second question, we introduce the variant of GO-Only (Global Optimization Only), in which the model relies exclusively on the global optimization, with $w_{local}$ in Equation 10 set to zero. Regarding the third question, we propose the variant of GO-Blended (Global Optimization Blended), which applies soft blending without any specific handling of all-zero cases. Additionally, we investigate the impact of the weight and threshold of soft blending on model performance. Table 3 presents the experimental results of these variants of COPO on the MATH-500 dataset.

**Effectiveness of "Ineffective" Data** Under the GO-Selective setting, the global optimization is utilized exclusively in cases where all sampled answers generated by the model are incorrect. The

GO-Selective experiment exclusively optimizes the fully incorrect paths that fail to receive effective advantage signals within GRPO, thereby providing targeted evidence of our method's ability to extract effective signals from "ineffective data" deprecated by DAPO. On the MATH-500 dataset, GO-Selective achieves improvements of 3.05% and 2.08% over the baseline in terms of the mean and maj metrics, respectively. This demonstrates that training data with all-zero outcomes still holds learning value, and the incorporation of global optimization enables the model to effectively leverage useful information from those fully incorrect training examples.

**Impact of Global Signals**  To evaluate whether the introduction of a global optimization mechanism leads to tangible performance improvements, we introduce the GO-Only experiment, in which the model is updated solely based on the advantage derived from the global reward. As shown in Table 3, the GO-Only setting achieves strong performance on both the mean@8 and maj@8 metrics, significantly outperforming the baseline with 4.52% improvements in mean@8 and 2.17% in maj@8, consistently outperforming the baseline. This result indicates that our global optimization formulation allows the model to capture both positive signals from correct trajectories and penalties from incorrect ones, thereby improving overall performance.

**Influence of the Hybrid Strategy**  Under the GO-Blended setting, the model achieves performance improvements of 3.97% and 1.89% on the mean@8 and maj@8 metrics compared to the baseline, demonstrating that our soft-blending approach effectively integrates the two optimization strategies. The method of combining the global optimization with the original local optimization in GRPO also leads to different impacts on the final results. As shown in Equation 10, higher consistency entropy of the answer list corresponding to greater weight assigned to the local loss, and lower entropy results in greater weight for the global loss. The weight allocation is controlled by the parameters $\gamma$ and $\rho$ in the equation.

Table 4: Ablation study of soft-blending weights $\gamma$ and $\rho$ on Qwen2.5-Instruct 3B.

| $\gamma$ | $\rho$ | MATH-500$^\dagger$ | MATH-500$^\ddagger$ |
|---|---|---|---|
| 3 | 1 | 55.18 | 60.81 |
| 5 | 1 | 59.05 | 63.75 |
| 10 | 1 | 59.40 | 63.97 |
| 20 | 0.5 | 56.23 | 61.97 |
| 20 | 1.2 | 59.30 | 64.12 |
| 20 | 1.5 | **60.38** | **65.06** |

The manner of integrating global loss with GRPO's original local loss also significantly influences performance. With an increasing slope, the weight distribution becomes more binary, indicating a preference for using either global or local optimization exclusively. In contrast, when the slope is smaller, the weight distribution tends to be more linear, suggesting that the loss computation incorporates both types of loss.

Table 4 shows that as the threshold $\gamma$ increases from 3 to 10, model accuracy on the benchmark gradually improves. This effect may arise from partial signal cancellation between the loss types, where the global term reduces inter-sample differences and weakens contrastive effectiveness.

In our soft-blending strategy, the proportion of global loss is controlled via the threshold parameter $\rho$ in Equation 10. With a smaller threshold, more trajectories upper to the threshold are assigned a high $w_{local}$, meaning a larger portion of the training data relies mainly on local rewards. Conversely, a higher threshold results in $w_{local}$ approaching zero, indicating a greater reliance on global loss.

We examined the impact of varying threshold $\rho$ values on model performance. The accuracy curves for GRPO and small thresholds ($\rho = 0.5$) show a declining trend in later training stages. As the threshold increases, the accuracy on the MATH-500 dataset improves progressively, suggesting that greater use of global optimization enhances the model's performance on mathematical reasoning tasks. From the above results, it can be observed that setting a larger slope $\gamma$ and a higher threshold $\rho$ in COPO training leads to better reasoning performance (e.g., $\gamma = 20$, $\rho = 1.5$). Lower parameter values, in contrast, result in diminished performance gains.

## 6 CONCLUSIONS

In this paper, we propose a novel consistency-aware policy optimization framework that incorporates a structured global reward mechanism based on outcome consistency, while employing an entropy-based soft-blending strategy to effectively integrate local and global optimization objectives. By effectively leveraging the information embedded in challenging training data, COPO achieves an important improvement over GRPO, suggesting that fully utilizing intra-group data with zero advantage values contributes positively to the training process. More details will be discussed in the appendix.

## 7 REPRODUCIBILITY STATEMENT

We have made every effort to ensure the reproducibility of our results. The code, developed on top of the VERL framework, has been anonymized and included in the supplementary materials. Detailed descriptions of the experimental setup, including model configurations and hardware specifications, are provided in section 4 and section A.2.

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

# A    APPENDIX

## A.1    ALGORITHM EXPLAINATION

### A.1.1    PROXIMAL POLICY OPTIMIZATION, PPO

The objective function for conventional PPO is defined as:

$$J_{\text{PPO}}(\theta) = \mathbb{E}_{q,o\sim\pi_{\theta_{\text{old}}}} \left[ \sum_{t=1}^{|o|} \min\left( \frac{\pi_\theta(o_t \mid q, o_{<t})}{\pi_{\theta_{\text{old}}}(o_t \mid q, o_{<t})} A_t, \, \text{clip}(r_t(\cdot)) A_t \right) \right] \tag{12}$$

where $\theta$ represents the parameters of the current policy $\pi_\theta$; $o_t$ is the token generated at step $t$, $o_{<t}$ represents the preceding token sequence. $A_t$ is the advantage function, which captures the relative value of taking action $o_t$ at state $s_t$ and is computed by $A_t = r_t + \gamma V(s_{t+1}) - V(s_t)$, where $V(s_t)$ is the value of state $s_t$ that is usually estimated by a value network. $\gamma \in [0, 1)$ is the discount factor, controlling the trade-off between immediate and future rewards. PPO uses the clipping operator:

$$\text{clip}(\cdot) = \text{clip}(r_t(\theta), 1 - \epsilon, 1 + \epsilon)$$

to restrict the update ratio $r_t(\theta)$ within the interval $[1 - \epsilon, 1 + \epsilon]$, and $\epsilon$ is the clip range of the importance sampling ratio. By doing so, PPO prevents excessively large policy updates that could destabilize training, ensuring that the new policy does not deviate too far from the previous one while still allowing sufficient flexibility for improvement.

### A.1.2    GROUP RELATIVE POLICY OPTIMIZATION, GRPO

GRPO simplifies the training process compared to PPO by utilizing reward-based advantage estimation. Instead of relying on a separate value network, GRPO calculates advantages by directly comparing rewards among samples generated from the same input, streamlining the overall architecture.

Given an input prompt q, the previous policy $\pi_{\theta_{\text{old}}}$ produces a set of G candidate output sequences, denoted as $\mathcal{O}q = \{o_1, o_2, \ldots, o_G\}$. Each sequence is subsequently evaluated using a task-specific reward function $r\phi$, constructed in accordance with the optimization objective, resulting in a corresponding reward set $\{r_1, r_2, \ldots, r_G\}$. The direction of the policy update is determined by the relative ranking of rewards within the group. Samples that receive rewards above the group average are encouraged by increasing their likelihood under the policy, while those with below-average rewards are discouraged by reducing their corresponding policy probabilities.

Based on this, the GRPO advantage is computed as:

$$\hat{A}_i = \frac{r_i - \mu_r}{\sigma_r},$$

where $\mu_r = \text{mean}(\{r_i\}_{i=1}^G)$, $\sigma_r = \mu_r = \text{std}(\{r_i\}_{i=1}^G)$. By substituting the new advantage $\hat{A}_i$, the group B, and the response set $\{o_1, o_2, \ldots, o_G\}$ sampled by the policy model into the PPO objective, we derive the objective function of GRPO:

$$J_{\text{GRPO}}(\theta) = \mathbb{E}_{q,\{o_i\}\sim\pi_{\theta_{\text{old}}}} \left[ \frac{1}{G} \sum_{i=1}^G \frac{1}{|o_i|} \sum_{t=1}^{|o_i|} \min\left( \frac{\pi_\theta(o_{i,t} \mid q, o_{i,<t})}{\pi_{\theta_{\text{old}}}(o_{i,t} \mid q, o_{i,<t})} \hat{A}_i, \, \text{clip}(\cdot) \hat{A}_i \right) - \beta \, \mathbb{D}\text{KL}\left[ \pi_\theta \parallel \pi_{\text{ref}} \right] \right].$$

### A.1.3    RULE-BASED REWARD

Rule-based reward assigns scores to model outputs based on predefined rules. In our setting, correctness is the only evaluation criterion, which helps reduce the risk of reward hacking. Specifically, the model is prompted to generate responses in a required format, and the final answer is extracted and directly compared with the ground truth to assign the reward:

$$R(o) = \begin{cases} 1, & \text{is\_equivalent}(\tau, \hat{\tau}) \\ 0, & \text{otherwise} \end{cases} \tag{13}$$

where $\tau$ is the predicted answer extracted from response $o$ and $\hat{\tau}$ is the ground truth.

### A.1.4 DEMONSTRATIVE EVALUATION OF COPO FRAMEWORK

**Demonstration of the COPO computation procedure**    To illustrate the operational mechanism of COPO, we present a concrete example. Consider a batch consisting of 5 data instances, where the model generates 6 candidate responses for each instance. For demonstration purposes, we take one example from the batch: ***the question "1 + 1 = ?"***. The model produces 6 reasoning-based responses to this question, such as: ***"The answer is 2. Answer: $2."*** From each response, the final predicted answer is extracted. Suppose the extracted answers are: **[2, 2, 2, 3, 3, 4]**. Based on ground truth comparison, the corresponding accuracy rewards are assigned as: **[1, 1, 1, 0, 0, 0]**. Subsequently, COPO computes the local rewards and local advantages for each response according to GRPO using 5:

$$\hat{A}_o^{\text{local}} = \frac{R(o) - \text{mean}\big(\{\, R(o_i) \,\}_{i=1}^G\big)}{\text{std}\big(\{\, \hat{R}(o_i) \,\}_{i=1}^G\big)}.$$

For this reward list, the mean is 0.5 and the standard deviation is 0.5, resulting in a local advantage list of **[1, 1, 1, -1, -1, -1]** for the corresponding sample. The final local loss is computed from this advantage using Equation 4:

$$J_{\text{local}}(\theta) = \mathbb{E}_{q,\{o_i\}\sim\pi_{\theta_{\text{old}}}}\left[\frac{1}{\sum_{i=1}^G |o_i|}\sum_{i=1}^G\sum_{t=1}^{|o_i|}\min\left(\frac{\pi_\theta(o_{i,t} \mid q, o_{i,<t})}{\pi_{\theta_{\text{old}}}(o_{i,t} \mid q, o_{i,<t})}\hat{A}_{o_i}^{\text{local}}, \text{clip}(\cdot)\hat{A}_{o_i}^{\text{local}}.\right)\right],$$

Next, based on the Equation:

$$\hat{R}(q) = \frac{1}{G}\sum_{i=1}^G r_i, \tag{14}$$

the global reward for this data prompt is calculated to be **0.5**. For each of the 5 samples in the batch, the global reward can be computed by following the procedure described above, resulting in 5 values. We set the global rewards for these samples as:

$$[\,\frac{1}{6},\ \frac{1}{6},\ \frac{2}{3},\ \frac{1}{2},\ \frac{1}{2}\,].$$

The global advantage is then calculated based on Equation 6:

$$\hat{A}_q^{\text{global}} = \frac{\hat{R}(q_j) - \text{mean}\big(\{\, \hat{R}(q_j) \,\}_{j=1}^B\big)}{\text{std}\big(\{\, \hat{R}(q_j) \,\}_{j=1}^B\big)}, for\ \forall\, o_i \in \mathcal{O}_q.$$

For this global reward list, the mean is 0.4 and the standard deviation is 0.2, resulting in a local advantage list of

$$[-1.167,\ -1.167,\ 1.333,\ 0.500,\ 0.500,\ 0.500]$$

for the corresponding sample. The final global loss is computed from this advantage using Equation 7.

$$J_{\text{global}}(\theta) = \mathbb{E}_{q,\{o_i\}\sim\pi_{\theta_{\text{old}}}}\left[\frac{1}{\sum_{i=1}^G |o_i|}\sum_{i=1}^G\sum_{t=1}^{|o_i|}\quad\min\left(\frac{\pi_\theta(o_{i,t} \mid q, o_{i,<t})}{\pi_{\theta_{\text{old}}}(o_{i,t} \mid q, o_{i,<t})}\hat{A}_q^{\text{global}}, \text{clip}(\cdot)\hat{A}_q^{\text{global}}\right)\right].$$

Subsequently, given the extracted answer list [2, 2, 2, 3, 3, 4] for this data prompt, the consistency entropy is calculated using Equation 8,

$$H(q) = -\sum_{\tau \in T_q} p(\tau) \cdot \log p(\tau),\ \ p(\tau) = \frac{\text{count}(\tau)}{G},$$

where $\text{count}(\tau)$ denotes the number of occurrences of $\tau$. For this example, we have:

$$p('2') = 0.5,\ p('3') = \frac{1}{3},\ p('4') = \frac{1}{6}.$$

The resulting consistency entropy $H$ is 1.459.

By substituting the consistency entropy into the weight computation formula (Equation 10), where we set $\gamma = 3$ and $\rho = 1$, the sigmoid function returns $w_{\text{local}} = 0.799$, and thus $w_{\text{global}} = 0.201$.

---

**System Prompt**

You are Qwen, created by Alibaba Cloud. You are a helpful assistant.

**User Prompt**

Solve the following math problem step by step. The last line of your response should be of the form Answer: $Answer (without quotes) where $Answer is the answer to the problem.
{Question}
Remember to put your answer on its own line after "Answer:".

---

Figure 4: Training and Test Prompts

$$\mathcal{L}_q = w_{\text{local}}(H(q)) \cdot \mathcal{L}_{\text{local}}(q) \; + \; w_{\text{global}}(H(q)) \cdot \mathcal{L}_{\text{global}}(q),$$

and the weighting functions are defined as:

$$w_{\text{local}}(H) = \sigma(\gamma(H - \rho)), w_{\text{global}}(H) = 1 - w_{\text{local}}(H).$$

Finally, according to Equation 3, the local loss and global loss are combined using $w_{\text{local}}$ and $w_{\text{global}}$ to obtain the final loss value.

$$J_{\text{COPO}}(\theta) = \mathbb{E}_{q \sim \mathcal{D}} \Big[ w(H_q) \cdot \mathcal{L}_{\text{local}}(q) + (1 - w(H_q)) \cdot \mathcal{L}_{\text{global}}(q) \Big].$$

**Utilization of Data with Zero Local Advantage in COPO**  COPO provides optimization signals for data with zero advantages in GRPO, thereby preventing gradient vanishing and sample wastage. Specifically, consider a batch with five data instances, whose corresponding accuracy reward lists are **[0, 0, 0, 0, 0, 0]**, **[0, 0, 0, 0, 0, 0]**, **[1, 1, 1, 1, 1, 1]**, **[0, 0, 0, 1, 1, 1]**, and **[0, 0, 0, 1, 1, 1]**. For the first three instances, since all responses receive uniform local rewards, their local advantages are zero according to Eq.5.

Without incorporating the global advantage, their final advantages remain zero, resulting in zero gradients and thus gradient vanishing. COPO addresses this issue by assigning global advantages to the data. According to Eq.14, the global rewards of the five samples are computed as the mean accuracy of the model's responses, yielding **[0, 0, 1, 0.5, 0.5]**. Based on Eq.7, the corresponding global advantages are **[-1.07, -1.07, 1.60, 0.27, 0.27]**, where the first three instances have nonzero global advantages that reflect the model's accuracy on these data. Finally, by Eq.3, these nonzero global advantages contribute to the total advantage of the first three data points, thereby avoiding gradient vanishing.

### A.1.5  REWARD HACKING IN MULTI-OBJECTIVE OPTIMIZATION

When applying multiple rewards for multi-objective optimization, advantage degeneration serves as a direct cause of reward hacking. When different reward signals have varying degrees of difficulty to achieve, the model tends to concentrate its strategy on optimizing the easier objective.

For example, when designing both a format reward and an outcome correctness reward, the initial policy finds it much easier to satisfy formatting requirements than to achieve correct reasoning. Consequently, the model rapidly shifts to producing outputs that conform to format specifications while ignoring reasoning quality. This leads to reward homogenization within the group, further degenerating the advantage estimation and causing the training process to collapse without further effective learning.

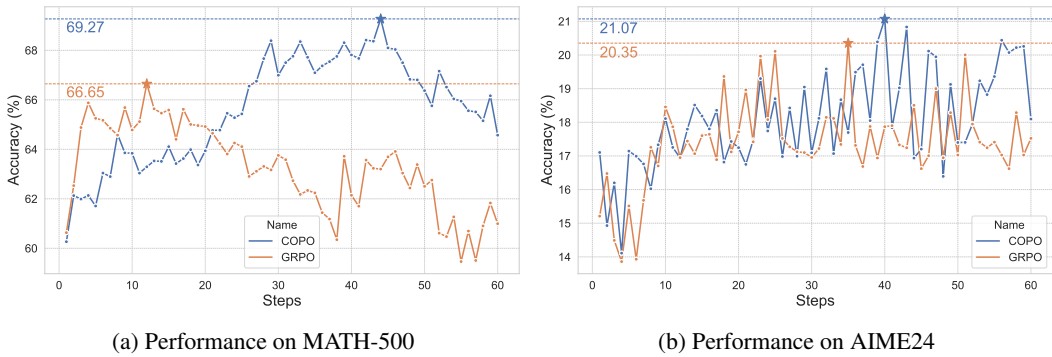

(a) Performance on MATH-500    (b) Performance on AIME24

Figure 5: Performance of GRPO and COPO on MATH-500 and AIME24 (maj@8) using Qwen2.5-7B-Instruct during training

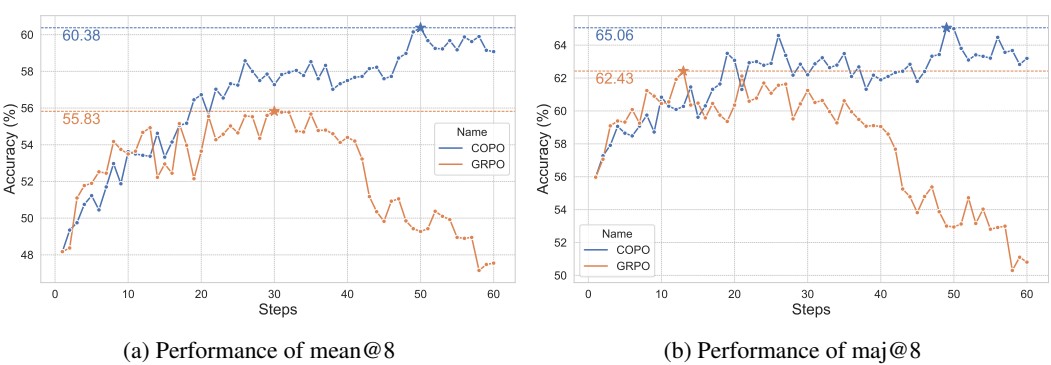

(a) Performance of mean@8    (b) Performance of maj@8

Figure 6: Performance of GRPO and COPO on MATH-500 with mean@8 and maj@8 using Qwen2.5-3B-Instruct during training

## A.2 EXPERIMENTS SETTING DETAILS

All experiments for the 3B model were conducted on four GPUs with 80 GB of memory each, while those for the 7B model were carried out on four GPUs with 96 GB of memory each. During evaluation, the dataset used the same prompts as the training set (DAPO-MATH-17k) to ensure consistency. Figure 4 presents the detailed structure of the prompt. When obtaining rule-based rewards, we extract the final answer from the reasoning path in the required format and use the Python package $math\_verify$ to determine whether the answer matches the ground truth.

## A.3 MORE EXPERIMENTS RESULTS

### A.3.1 FIGURES OF MAIN EXPERIMENTS

The test performance of Qwen2.5-7B-Instruct under the GRPO and COPO algorithms is compared in Figure 5. The progression of maj performance over the course of training is shown in subfigures (a) and (b) for the MATH-500 and AIME24 datasets, respectively. COPO demonstrates more consistent gains in maj accuracy over GRPO on both datasets, suggesting that it enables the model to acquire more general and transferable problem-solving strategies.

The test performance of Qwen2.5-3B-Instruct under the GRPO and COPO algorithms on MATH-500 is compared in Figure 6. The COPO method demonstrates a consistent upward trend in both the mean@8 and maj@8 metrics.

Figure 7 shows the entropy dynamics of the 7B and 3B models during training with COPO and GRPO. For the 7B model, the entropy trends of COPO and GRPO are similar, but COPO maintains a more stable entropy level in the later steps. For the 3B model, COPO yields consistently higher entropy, indicating its ability to preserve response diversity throughout training.

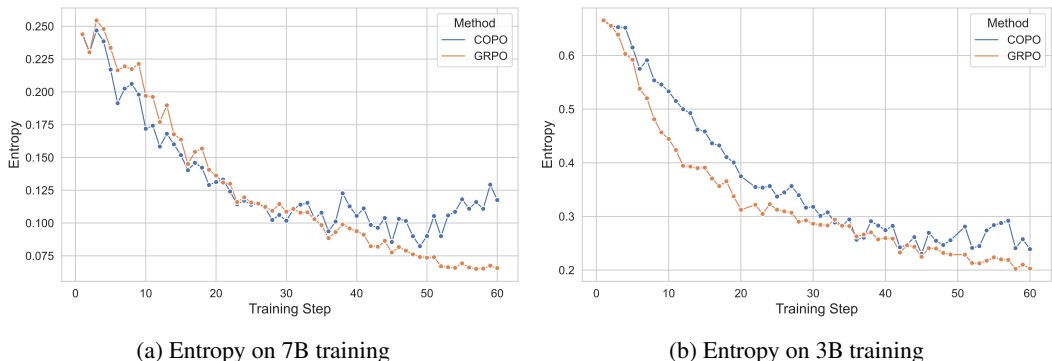

(a) Entropy on 7B training             (b) Entropy on 3B training

Figure 7: Entropy of COPO training on Qwen2.5-7B-instruct and Qwen2.5-3B-Instruct

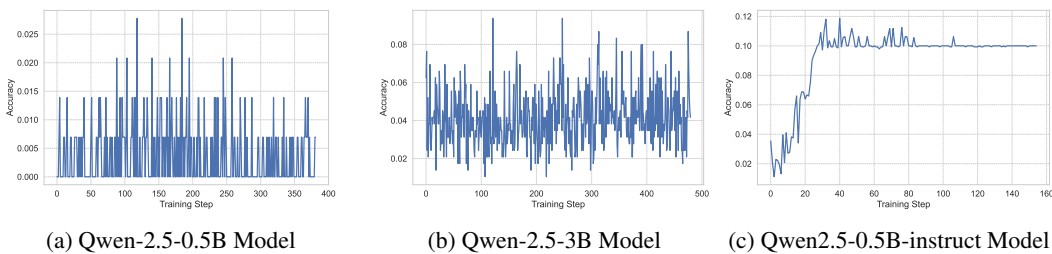

(a) Qwen-2.5-0.5B Model      (b) Qwen-2.5-3B Model      (c) Qwen2.5-0.5B-instruct Model

Figure 8: Training Accuracy Dynamics of GRPO on the Small and Base Models

### A.3.2 EXPERIMENTS ON SMALL MODEL AND BASE MODEL

We conducted GRPO experiments on base models (Qwen2.5-0.5B, Qwen2.5-3B) as well. Due to their lack of instruction-following ability, these models struggled to produce correctly formatted outputs. To address this, we introduced a format reward:

$$R(\tau, \hat{\tau}) = \begin{cases} 0, & \text{is\_null}(\hat{\tau}) \\ 1, & \text{is\_equivalent}(\tau, \hat{\tau}) \\ 0.1, & \text{otherwise} \end{cases} \tag{15}$$

where $\tau$ is the formatted answer extracted from prediction and $\hat{\tau}$ is the ground-truth. The format reward is defined as 0 for incorrect formats, 0.1 for correct format but incorrect answers, and 1 for correct answers.

However, even with the format reward, the models failed to maintain proper output formatting. As shown in Figure 8a and 8b, the reward score remained below 0.1 after 300 training steps with no upward trend.

We also ran experiments on Qwen2.5-0.5B-Instruct model, applying the same format reward to regulate output. According to Figure 8c, under this scheme, the model achieved stable formatting within 40 steps. However, due limited base capabilities, it was unable to sample correct answers on this dataset, with most prompts yielding zero advantage and no further learning progress.

These results suggest that RL methods cannot directly drive small models that lack instruction-following ability toward desired behaviors. Dataset difficulty calibration and cold-start strategies may be necessary prerequisites for RL training on small base models.

### A.3.3 IMPACT OF LOSS MASKING ON FULLY INCORRECT SAMPLES

To further investigate whether fully incorrect samples contribute to model learning, we conducted an additional experiment called COPO-Selective, in which the loss corresponding to fully incorrect samples is set to zero, while the remaining samples still use soft blending to combine local and global losses. Compared to our main method, the only difference in COPO-Selective is how fully incorrect samples are handled. The main method applies global loss to these samples, while COPO-

Table 5: Comparison of COPO-Selective and other methods across MATH-500 datasets.

| Method | loss for all-zero | soft blending | MATH-500 mean@8 | MATH-500 maj@8 |
|---|---|---|---|---|
| GRPO | zero | ✗ | 55.83 | 62.43 |
| COPO-Selective | zero | ✓ | 59.18 | 64.57 |
| COPO | global loss | ✓ | 60.38 | 65.06 |

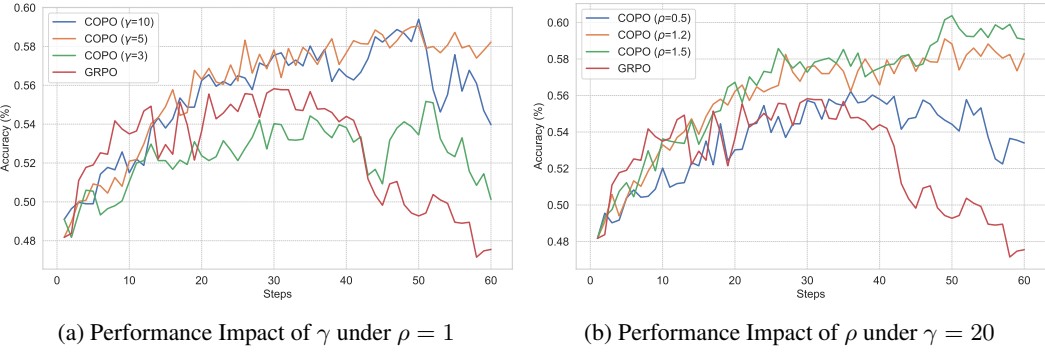

(a) Impact of $\gamma$ on $w_{\text{local}}$ under $\rho = 1$    (b) Impact of $\rho$ on $w_{\text{local}}$ under $\gamma = 10$

Figure 9: Variation of $w_{\text{local}}$ with Consistency Entropy $H$ under Different Hyperparameter Settings

Selective excludes them from optimization by assigning a zero loss, effectively removing them from weight updates.

As shown in Table 5, COPO-Selective achieves a significant improvement over GRPO, but still underperforms the main method by 1.2% and 0.49% in mean@8 and maj@8, respectively. This suggests that incorporating loss signals for fully incorrect samples with zero intra-group advantage helps the model extract useful information from them.

## A.4 DISCUSSION

### A.4.1 EFFECT OF HYPERPARAMETERS IN SOFT BLENDING

Figure 9 illustrates the effect of the soft-blending hyperparameters $\gamma$ and $\rho$ on the weight $w_{\text{local}}$. From subfigure (a), we observe that with $\rho$ fixed, $\gamma$ controls the sharpness of the soft blending curve. As $\gamma$ increases, the curve transitions from linear to more binary-like, meaning that larger values of $\gamma$ push $w_{\text{local}}$ closer to 0 or 1 for more data points. Subfigure (b) shows that with $\gamma$ fixed, $\rho$ determines the horizontal shift of the blending curve. Smaller values of $\rho$ shift the curve leftward toward the y-axis, resulting in more data points receiving $w_{\text{local}}$ values close to 1. In contrast, larger values of $\rho$ shift the curve rightward, assigning more data points with $w_{\text{local}}$ values close to 0, which indicates a greater reliance on global optimization.

Figure 10 shows how the performance of COPO on the MATH-500 test set varies under different hyperparameter settings. It can be observed that higher values of $\gamma$ and $\rho$ result in the highest

(a) Performance Impact of $\gamma$ under $\rho = 1$    (b) Performance Impact of $\rho$ under $\gamma = 20$

Figure 10: Effect of different $\gamma$ and $\rho$ in Soft Blending with Qwen2.5-3B-Instruct on MATH-500

**Question:**

Twelve friends met for dinner at Oscar's Overstuffed Oyster House, and each ordered one meal. The portions were so large, there was enough food for 18 people. If they share, how many meals should they have ordered to have just enough food for the 12 of them?

**GRPO:**

To solve this problem, we need to determine how many meals the 12 friends should have ordered so that they have just enough food for themselves, given that the portions were enough for 18 people.

First, we need to find out how much food each person consumes. Since the portions are large and are enough for 18 people, we can say that the total portions of food for 18 people are enough for each of the 18 people to eat a full meal. Therefore, the food for one person is $\frac{1}{18}$ of the total food.

Now, we need to find out how much food is enough for 12 people. Since each person requires $\frac{1}{18}$ of the total food, for 12 people, the total amount of food needed would be:

$$12 \times \frac{1}{18} = \frac{12}{18} = \frac{2}{3}$$

This means that to have just enough food for 12 people, they should have ordered $\frac{2}{3}$ of the total portions. Since the portions they had were enough for 18 people, and they need just enough for 12 people, they should have ordered the equivalent of what 6 people would have ordered (since $\frac{2}{3}$ of 18 people is 6 people).

Therefore, they should have ordered meals for 6 people to have just enough food for 12 people.

So, the answer is:

**Answer: $6 →Wrong**

**COPO:**

To solve this problem, we need to determine how many meals each person should have ordered so that the total amount of food is just enough for 12 people, given that the original portions are enough for 18 people.

First, let's determine the ratio of the number of people the original meals are enough for to the number of people they want to serve. The original meals are enough for 18 people, and they want to serve 12 people. The ratio of the desired number of people to the original number of people is:

$$\frac{12}{18} = \frac{2}{3}$$

This means that each person should receive $\frac{2}{3}$ of a meal to serve 12 people with the amount of food originally intended for 18 people.

Since each of the 12 friends should receive $\frac{2}{3}$ of a meal, the total number of meals needed is:

$$12 \times \frac{2}{3} = 8$$

Therefore, they should have ordered 8 meals to have just enough food for the 12 of them.

**Answer: $8 →Right**

Figure 11: Case of GRPO and COPO reasoning results on MATH-500

accuracy. This suggests that a more binary-like blending curve, along with a greater reliance on global optimization, can more effectively improve model performance.

## A.5 CASE STUDY

To better understand the difference between methods, we conduct a case study on selected examples from the MATH-500 dataset. Figure 11 illustrates a representative example where GRPO fails due to incorrect intermediate reasoning, while COPO provides a complete and correct derivation.

## A.6 LIMITATION

As shown in Table 1, when using the relatively small 3B model, our method exhibits weaker performance on AIME24, with a difference of 0.41% compared to the baseline. However, it achieves greater improvements on the simpler MATH-500 dataset. We also conducted experiments with the

Table 6: Comparison of GRPO and our method across MATH-500 and AIME24 datasets.* denotes the results are reproduced by ourselves

| Method | MATH 500 | | AIME 24 | | Mean Avg | Maj Avg |
| | mean@8 | maj@8 | mean@64 | maj@64 | | |
|---|---|---|---|---|---|---|
| Qwen2.5-instruct 1.5B* | 66.88 | 71.00 | 8.80 | 18.14 | 63.59 | 68.01 |
| GRPO | **70.00** | **73.55** | **11.46** | 19.23 | **66.69** | **70.48** |
| COPO ($\gamma = 5$, $\rho = 1$) | 68.93 | 72.85 | 10.78 | 19.46 | 65.64 | 69.83 |
| COPO ($\gamma = 10$, $\rho = 1$) | 68.83 | 73.12 | 10.78 | **19.92** | 65.54 | 70.11 |

COPO method on the Qwen2.5-Math-1.5B-Instruct model. Table 6 presents a performance comparison between our method and the baseline on the MATH-500 and AIME24 datasets, using both the mean and maj metrics. As shown, our method still lags behind GRPO by approximately 1% on most metrics.

This observation suggests that the current COPO method may not offer advantages when applied to smaller math-tuned models. On one hand, smaller models typically have weaker generalization capabilities, making it difficult to fully leverage the potential benefits of combining local and global losses. In some cases, the objectives of local and global optimization may even conflict, leading to degraded performance. On the other hand, the Qwen2.5-Math-1.5B-Instruct model is specifically fine-tuned for mathematical tasks. Introducing a composite loss function that is not fully aligned with its task-specific pretraining objectives may interfere with its learned structural representations or reasoning mechanisms, thereby weakening overall performance.

## A.7 RELATED WORKS

### A.7.1 LLM REASONING

The ability of LLMs to directly generate answers through autoregressive decoding is often referred to as their 'System 1' capability Li et al. (2025). In contrast, solving complex problems through deliberate, logical reasoning—by first thinking and then generating—is considered the 'System 2' mode. CoT prompting has emerged as one of the most effective approaches to endow LLMs with human-like reasoning ability. Early CoT Wei et al. (2022); Jiang et al. (2025) methods relied on in-context learning by inserting exemplar reasoning processes into prompts, but such methods struggle to generalize across a wider range of task domains. An alternative and more scalable approach is to let models autonomously generate reasoning paths depending on the specific question. By fine-tuning LLMs on high-quality reasoning trajectories, models can quickly learn human-like thought patterns for particular problems. However, the annotation cost of such data is often prohibitive for most researchers. As a result, a series of RL-based methods have emerged to improve the reasoning abilities of LLMs without requiring fully supervised data.

### A.7.2 RL-BASED POSTED-TRAINING

Early RL-based post-training methods focused primarily on aligning model outputs with human preferences in multiple dimensions such as non-toxicity, fairness, or politeness, rather than explicitly enhancing reasoning capability. The release of OpenAI's O1 Jaech et al. (2024) model shifted attention toward improving reasoning via Monte Carlo Tree Search (MCTS) and process-level rewards, encouraging models to explore higher-quality reasoning trajectories. However, this approach still requires extensive computational resources to supervise the exploration process and provide reward or value signals. DeepseekMATH Shao et al. (2024b) introduced the GRPO training method and demonstrated that sparse, outcome-level rewards could also guide models toward discovering correct reasoning paths. R1 further proposed a rule-based reward system, removing the need for a learned reward model and reducing computational overhead. Nevertheless, the inherent limitations of GRPO led to the frequent disappearance of optimization signals within groups. DAPO Yu et al. (2025) attempted to address instability and inefficiency during training, but it did not fundamentally resolve the sample inefficiency caused by the design of GRPO.

## A.8 LLM Usage

During manuscript preparation, Large Language Models (LLMs) were employed **solely** for linguistic refinement. Their use was restricted to improving grammar, readability, and stylistic clarity, without any involvement in research conception, methodology, experimental design, data analysis, or interpretation of results. All scientific ideas, analyses, and conclusions are entirely the work of the authors. The LLM-assisted edits were carefully reviewed to ensure accuracy, originality, and adherence to ethical standards. The authors retain full responsibility for the content of this manuscript.

