# OpenReview forum: "COPO: Consistency-Aware Policy Optimization"
_ICLR.cc/2026/Conference — ICLR 2026 Conference Withdrawn Submission_

### Official Review · Reviewer_oagU · 2025-11-01

**Soundness:** 2
**Presentation:** 2
**Contribution:** 2
**Rating:** 4
**Confidence:** 4

**Summary:**

This paper addresses a failure mode in rule-based RL for LLM reasoning where multiple samples per prompt collapse to the same outcome, driving group-based advantages to zero and stalling learning. It proposes Consistency-Aware Policy Optimization: a structured global reward tied to outcome consistency and a corresponding global loss that preserves gradients even under high agreement, encouraging correct, self-consistent reasoning. An entropy-based soft blending adaptively mixes local advantage estimates with the global objective to balance exploration and convergence.

**Strengths:**

- The paper does a good job of identifying, explaining, and quantifying the "advantage degeneration" problem in GRPO, showing it affects ~60% of samples for the Qwen2.5-3B-instruct model.
- The core idea of introducing a batch-level "global" advantage to provide a signal when the "local" advantage vanishes is an elegant and effective solution. It correctly identifies the wastefulness of simply discarding this data.
- The use of consistency entropy to soft-blend the local and global losses is an intelligent addition, allowing the model to rely on the most appropriate signal dynamically.

**Weaknesses:**

- The primary weakness of this paper is its limited novelty. The proposed method is a highly incremental design on top of GRPO, essentially adding a "global" reward layer to the existing "local" one. This combination of high complexity for a small incremental improvement feels more like an engineering design than a fundamental contribution.
- The method does not fundamentally solve the "zero advantage" problem. The paper claims to address "advantage degeneration", but it merely sidesteps the issue by adding a separate, parallel optimization signal (the global loss). It does not resolve the underlying cause of the local advantage collapse, and this claim lacks strong empirical support.
- The paper's clarity and presentation need improvement. The issues begin with the title, which is abstract and not intuitive about the method's actual mechanism. The introduction is overly brief, which fails to sufficiently motivate the problem or clearly articulate the core contribution, making the paper difficult to follow. In Table 3, the GO-only approach with the best performance actually did not address this issue.
- The ablation study in Table 3 shows that the "GO-Only" variant achieves a 60.35 mean@8 score, while the full, complex COPO method achieves 60.38. This negligible 0.03% difference suggests the primary benefit comes from the global optimization, and the complex, hyperparameter-sensitive entropy blending adds little to no value.
- The method's effectiveness is sensitive to model scale and task. The authors honestly report that COPO underperforms the GRPO baseline on the 3B model for AIME24 and on the 1.5B math-tuned model. This suggests the proposed global optimization may conflict with smaller models' capacities or specialized tuning.

**Questions:**

- The "GO-Only" ablation in Table 3 achieves 60.35 mean@8, while the full COPO method (which includes soft-blending) achieves 60.38. Given this negligible difference, can the author elaborate on the specific benefits? Why not simply simplify the framework of this paper to "GO-Only"?
- The performance degradation on the Qwen2.5-Math-1.5B-Instruct model is a concern. Could the authors elaborate on their hypothesis that this is a "conflict" with its pretraining?
- Your ablation in Table 2 suggests that token-level loss is not beneficial. The DAPO claims it is a key beneficial component. How do you explain this direct contradiction?
- The 59.9% of "ineffective" samples include 56.0% all-incorrect and 3.9% all-correct. The "GO-Selective" ablation was only tested by applying global loss to the entire incorrect group. Is there also a benefit to applying global loss to the all-correct group? By the way, the author has handled the situation where all cases fail to avoid failure in "entropy-based" blending. But what is the situation of full success?

---

### Official Review · Reviewer_czru · 2025-11-03

**Soundness:** 2
**Presentation:** 2
**Contribution:** 2
**Rating:** 2
**Confidence:** 4

**Summary:**

The paper proposes COPO. It is a GRPO-style method for LLM reasoning. It has two main ideas. First, it uses a batch-level global advantage. This advantage standardizes the mean reward for prompts in a minibatch (Eq. 6). Second, it use entropy for a soft blend. This blends local and global losses (Eq. 9 to 11). The entropy comes from the G rollouts' final answers. Tests on math benchmarks (MATH-500, AIME-24/25, GSM8K) used Qwen2.5-3B/7B. The results show small gains over GRPO and DAPO. The best gains were 2–4% on MATH-500. But, it showed some regressions on AIME and a 1.5B model.

**Strengths:**

The paper identifies a GRPO failure mode. This happens when all *G* rollouts for a prompt agree. They are all-right or all-wrong. This agreement creates near-zero standardized advantages. Which wastes samples. Fig. 1 shows this happens often in the 3B setup.

**Weaknesses:**

1.  The global advantage are the prompt-level mean reward. It is standardized by the minibatch mean/std (Eq. 6). This makes a constant baseline for all trajectories of a prompt. This is a standard variance-reduction control-variate. Calling it a "global optimization mechanism" overstates its novelty. Eq. 7 applies this same advantage to all tokens and trajectories. This discards the intra-prompt credit assignment. GRPO was designed to keep that assignment.

2.  The paper claims COPO fixes gradient-vanishing (Abstract). It gives intuition, not proofs. There is no unbiasedness or varience analysis for the entropy gate. There is no convergence argument. The text says the batch statistics "do not introduce bias" because they are treated as constants but the algorithm does not specify stop-gradient/detach for the batch mean/std. This detail is needed for unbiased policy-gradient estimation.

3.  H(q) is computed from unique final answers only. It ignores the reasoning traces. Prompts with diverse wrong reasoning get high H. They get local loss. Prompts with uniform correct reasoning get low H. They get global loss. This can reward superficial agreement. It penalize exploratory steps. The paper does not assess H(q). It does not check H(q) correlation with problem difficulty or verifier confidence.

**Questions:**

1.  Do you detach the batch mean/std in Eq. (6) during backprop? If not, what is the exact estimator? What are its bias and variance properties? Please provide a derivation.

2.  How do results change with *G* in {2, 4, 8}? How do they change with temperature/top-p sweeps and batch size *B*? The Fig. 1 degeneracy depends on these knobs. it needs stress-testing.

3.  Please compare to REINFORCE with a learned value head. Also compare to batch-normalized REINFORCE without the entropy gate. This will test if the batch baseline alone causes COPO's gaines.

---

### Official Review · Reviewer_CVzE · 2025-11-04

**Soundness:** 2
**Presentation:** 1
**Contribution:** 2
**Rating:** 4
**Confidence:** 3

**Summary:**

The paper tackles a familiar GRPO failure mode: when all rollouts for a prompt have the same outcome (all-correct or all-incorrect), intra-group reward variance collapses and the GRPO advantage goes to ~0, wasting those samples. COPO augments GRPO with (i) a batch-level, prompt-wise global advantage (standardized across the mini-batch) so zero-variance prompts still provide a learning signal, and (ii) an entropy-based soft gate that blends the local GRPO loss with the global loss based on the prompt’s consistency entropy (diversity of extracted answers). Using correctness-only rewards, the authors train Qwen2.5-Instruct (3B/7B) for ~60 steps on DAPO-MATH-17k with 512 prompts/batch and G=6 rollouts. On MATH-500 and AIME24 they report modest, consistent gains over GRPO on 7B; on 3B, COPO improves MATH-500 and is slightly worse on AIME24. Ablations show a strong global-only baseline, plus sensitivity to the gate’s parameters.

Disclosure: I used assistive writing tools to draft this review; all evaluations and judgments are my own.

**Strengths:**

Motivation matches a real pain point. GRPO’s zero-variance groups are common and wasteful; adding a batch-normalized prompt-level signal is a focused fix.

Drop-in practicality. The method is easy to implement atop GRPO: compute prompt returns, standardize across the mini-batch, and mix with a simple entropy gate.

Some positive results. On a stronger backbone (7B), COPO shows consistent improvements on math reasoning benchmarks with reasonably stable training curves.

Useful ablations. Global-only vs soft-blended vs selective, plus γ/ρ sweeps, help tease apart where the gate matters.

Clear recipe. Data/rollouts/steps, optimizer, clipping, and the minimal reward design are all specified, which helps reproducibility.

**Weaknesses:**

Incremental novelty. Batch-level baselines/standardization are classic variance-reduction ideas in policy gradients; here they’re applied at the prompt level and gated. Useful in practice, but not conceptually deep.

Benchmark choice feels dated. MATH-500 and GSM8K are saturated in 2025. The paper needs stronger generalization tests (recent AIME splits, IMO-style sets, and especially Putnam-AXIOM functional variations) to make the case.

Statistics are thin. Headline tables/curves lack confidence intervals, multi-seed averages, and significance testing. Given the small absolute gains, this is essential.

Forgetting not ruled out. Late-stage drops under RL can be catastrophic forgetting, not just “entropy collapse.” There are no retention or calibration diagnostics to separate these effects.

Non-standard metrics. The paper emphasizes mean@k/maj@k; Pass@k (at least Pass@1 and Pass@5) is expected in 2025 for comparability.

Presentation. The most important evidence appears late. Readers shouldn’t have to hunt for the main improvements; one decisive figure/table should be on page 1.

**Questions:**

Specific requests / experiments that would change my mind:

Harder generalization. Add Putnam-AXIOM functional variations (and recent AIME/IMO-style splits) with 95% CIs and p-values; show COPO beats strong GRPO/DAPO-style baselines at matched compute. I'd definitively increase my score if it increases performance on Putnam-AXIOM functional variance.

Pass@k + statistics. Provide Pass@1/Pass@5 with ≥5 seeds, paired-bootstrap CIs, and exact tests; include standardized effect sizes.

Checkpoint selection. Avoid cherry-picking the single “best” run; choose the best validation-significant checkpoint, then evaluate once on test.

Prompt-diversity control. Include a well-tuned prompt-only diversity baseline (explicitly enumerate multiple solution paths + probability assignment) to test whether RL is necessary for the observed gains. This is known to give diverse outputs.

---

### Note · Authors · 2025-11-25

I have read and agree with the venue's withdrawal policy on behalf of myself and my co-authors.